# Effect of Er,Cr:YSGG Laser Irradiation on the Surface Modification and Cell Adhesion on Titanium Discs: An In Vitro Study

**DOI:** 10.3390/ma17194899

**Published:** 2024-10-06

**Authors:** Takahiko Shiba, Kailing Ho, Xuehao Ma, Ye Won Cho, Chia-Yu Chen, David M. Kim

**Affiliations:** Division of Periodontology, Department of Oral Medicine, Infection and Immunity, Harvard School of Dental Medicine, Boston, MA 02115, USA; shibperi@gmail.com (T.S.); kailing_ho@hsdm.harvard.edu (K.H.); xuehao_ma@hsdm.harvard.edu (X.M.); yewon_cho@hsdm.harvard.edu (Y.W.C.); chia-yu_chen@hsdm.harvard.edu (C.-Y.C.)

**Keywords:** Er,Cr:YSGG, peri-implantitis, dental implant, machined surface, rough surface, titanium discs

## Abstract

This study evaluates the potential of erbium, chromium-doped:yttrium, scandium, gallium, and garnet (Er,Cr:YSGG) laser irradiation to modify the titanium surface for optimal seeding of fibroblasts and osteoblasts in the treatment of peri-implantitis. Titanium discs were treated using the Er,Cr:YSGG laser, an ultrasonic device with a stainless tip, or titanium scalers. Changes in surface properties were analyzed by profilometer and scanning electron microscopy (SEM). Murine fibroblast and osteoblast adhesion and proliferation were evaluated qualitatively and quantitatively at 24 and 72 h. Profilometric surface topography and SEM showed that titanium scalers and ultrasonic debridement techniques significantly changed the structure of the machined and rough titanium surfaces. The Er,Cr:YSGG laser irradiation, on the other hand, did not alter titanium microstructures. The Er,Cr:YSGG laser irradiation with the 40 Hz group showed a significantly higher attached fibroblast cell numbers than the titanium scaler group at 72 h after treatment (*p* = 0.023). Additionally, the number of the attached osteoblasts in the Er,Cr:YSGG laser irradiation with the 40 Hz group was significantly higher than that of the no-treatment groups 24 h after treatment (*p* = 0.045). The Er,Cr:YSGG laser effectively promoted adherence of fibroblasts and osteoblasts to the titanium surface without significantly altering the titanium surface, suggesting its superiority for treating peri-implantitis.

## 1. Introduction

In recent years, the utilization of dental implants for the restoration of missing or damaged teeth has significantly risen [1]. However, these dental implant treatments led to an increase in biological and prosthetic complications, such as infections, nerve damage, and implant fractures [2]. Among these complications, peri-implant mucositis and peri-implantitis, characterized by plaque-induced inflammation and rapid destruction of surrounding soft and hard tissues, occur at high incidence rates of 42.9% and 21.7%, respectively [3].

The treatment of peri-implant diseases requires a different approach from treating natural teeth, primarily due to the presence of implant threads that directly integrate with the surrounding alveolar bone. Successful treatment of peri-implant diseases relies on effective decontamination of bacterial biofilm from the implant surface [4]. Indeed, due to the difficulty in achieving complete decontamination, the successful re-establishment of osseointegration remains a significant challenge. de Waal et al. reported that clinical treatments for peri-implantitis, including those used for periodontal disease, were often ineffective [5]. In addition to implant macro- and micro-geometry, various intricate titanium implant surface topography designs differentially influence interactions with the peri-implant microenvironment. Rougher surfaces showed superior osteoblast attachment compared to smoother surfaces [6]. However, greater surface roughness facilitates greater biofilm accumulation and poses a challenge for decontamination [7].

Different mechanical and chemical decontamination approaches, such as the usage of ultrasonic scalers, titanium curettes, plastic curettes, erbium-doped yttrium aluminium garnet (Er:YAG) laser, and erbium, chromium-doped:yttrium, scandium, gallium, and garnet (Er,Cr:YSGG) laser have been proposed for peri-implantitis treatment, used either as monotherapy or as adjunctive therapy [8,9,10,11,12,13,14,15]. A study evaluating implant surface debridement method using various instruments showed that the Er:YAG and the Er,Cr:YSGG lasers offered greater advantages in eliminating calcified deposits on the microstructured surface of titanium implants without causing damage, compared to mechanical therapy using cotton pellets or titanium curettes [16]. Compared to Er:YAG, there are a limited number of studies on the effects on implant surfaces using the Er,Cr:YSGG laser, which has a different laser wavelength [17]. The effectiveness of the Er,Cr:YSGG laser for decontaminating and inducing surface changes is associated with factors such as power, frequency, pulse duration, tip diameter, and the distance between the tips and irradiated surfaces [18]. Chegeni et al. compared the proper irradiation conditions to avoid damage to the titanium implant surface using various conditions composed of 1.5 W/30 Hz or 2.5 W/30 Hz with conical, side-firing, and cylindrical tips. The titanium implant surface was not damaged when using side-firing tips at 1.5 W/30 Hz [19]. In another study, 2.5 W/25 Hz using a cylindrical tip was used to remove bacteria from titanium discs. The result showed the successful elimination of bacteria without surface alterations [20]. Meanwhile, Yao et al. indicated that treatment of titanium discs with the Er,Cr:YSGG laser at 1.5 W/20 Hz renders the discs slightly smoother, leading to a decrease in *Porphyromonas gingivalis* adhesion and an increase in fibroblast viability and osteoblast differentiation [21]. Strever et al. used the Er,Cr:YSGG laser at a power setting at 0 W, 0.5 W, 1 W, and 1.5 W with 30 Hz and a radial firing tip, which led to power-dependent ablation of *P. gingivalis* biofilm from titanium surfaces, without inducing measurable changes in temperature, surface microroughness, or water contact angle. This experiment needed a power setting of 1.0 to 1.5 W to ablate the bacteria coating the discs [22]. Irradiation conditions used in experiments varied depending on the report. Only a few papers reported the effect of the Er,Cr:YSGG laser on both machined and rough surfaces. Furthermore, to the best of our knowledge, no publications have concurrently assessed the proliferation rates of fibroblasts and osteoblasts after laser irradiation of both surfaces compared to other methods.

It is essential to consider appropriate laser irradiation conditions for treating peri-implantitis on machined and rough surfaces and for cell proliferation of fibroblast and osteoblast after the treatment. The purpose of the present study was (a) to compare the Er,Cr:YSGG laser-induced changes in machined and rough titanium surfaces with mechanical methods; and (b) to evaluate the effect of the Er,Cr:YSGG laser on fibroblast and osteoblast cell attachment and morphology on machined surface and rough surface titanium discs.

## 2. Materials and Methods

### 2.1. Titanium Disc Preparation

Machined and rough surface titanium discs were obtained from Hoowon EDI (Gimhae-si, Republic of Korea). Each disc measured 10 mm in diameter and 5 mm in thickness. The rough surface character was sand-blasted large-grit acid-etched.

#### Surface Treatment Methods

One control group (no treatment) and four treatment groups were formed for machined- and rough-surface titanium discs:No treatment (control group).Hand instrumentation was performed with a titanium scaler (Brasseler USA, Savannah, GA, USA) using 20 strokes on each disc.Ultrasonic instrumentation was performed using an ultrasonic scaler (Cavitron Plus Ultrasonic Scaler, Dentsply Sirona, York, PA, USA) with 20 strokes on each disc [23].Laser-treated instrumentation was performed using an Er,Cr:YSGG laser (Waterlase iPlus, Biolase, Irvine, CA, USA) with a wavelength of 2780 nm in short pulse “H” mode and 60 μs using a radial firing tip (RFPT 5) with a spot size of 1 mm. The power setting was 1.5 W with an air/water ratio of 40%/50%. The pulse rate was 30 Hz [23]. The laser tip was affixed to a handle and manually moved back and forth across the surface while maintaining a constant speed and distance of about 0.5 mm between the tip and the surface.In this laser-treated group, the pulse irradiation condition of group 4 was changed to 40 Hz, while all other conditions remained the same.

### 2.2. Surface Characterization

The post-treatment surfaces were cleaned with sterile water and photographed with a digital camera (Canon EOS 60D, Canon Inc, Tokyo, Japan). Additionally, the topography and surface roughness measurements of three discs in each group were conducted by a non-contact optical profiler (CCI MP, Taylor Hobson, Leicester, UK) with xy mode and “Sooth or Stepped” mode [24]. The scan range was set to 2.56 µm, with each scan completed in 1 s. The scanned area was 1.66 mm × 1.66 mm, and the data were collected at a resolution of 1024 × 1024 pixels. 3D topographic data were obtained. The parameters used for surface roughness measurement were squared mean height (Sq), arithmetic mean height (Sa), and the areal material ration (Smr). Three titanium discs were used in each group of statistical analyses, and three measurements were taken for the analysis of one disc, and the average of the measurements was used. Scanning electron microscopy (SEM) (Zeiss Gemini 360 FE-SEM SEC; ZEISS, Oberkochen, Germany) analysis was completed to examine any noticeable change in surface texture using one disc in each group at 5 kV. Secondary electron images (SE2) were obtained under high vacuum mode.

### 2.3. Fibroblast and Osteoblast Cell Adhesion

The post-treatment surfaces were cleaned with sterile water followed by sterile phosphate-buffered saline (PBS). Three discs in each group for two different cell types were used for quantitative analysis with a cell count using a luminescent cell viability kit (CellTiter-Glo Luminescent Cell Viability Assay, Promega, Madison, WI, USA), and one disc from each group was used for SEM qualitative analysis. Mouse fibroblast NIH3T3 cells were maintained in DMEM medium (Invitrogen; Thermo Fisher Scientific, Inc., Waltham, MA, USA) supplemented with 10% fetal bovine serum (FBS) (Sigma, St. Louis, MO, USA), 100 units/mL penicillin, and 100 μg/μL streptomycin at 37 °C. The cells were incubated in 5% carbon dioxide and 95% atmospheric air [23]. Mouse preosteoblast MC3T3-E1 subclone 14 cells were cultured in α-MEM (Thermo Fisher Scientific, Inc.) supplemented with 10% FBS, 100 units/mL penicillin, and 100 μg/μL streptomycin at 37 °C in 5% carbon dioxide and 95% atmospheric air [25]. Cells were seeded on the discs at 50,000 cells/cm^2^ in 1 mL of culture medium in each well of a 24-well plate for 24 h and 72 h following the completion of disc treatment. At the end of the incubation period, specimens were transferred to new wells, and unbound cells were washed off with PBS. Subsequently, a 500-μL mixture of the corresponding culture medium and CellTiterGlo solution at a 1:1 ratio was added to each well. Cell lysis was induced by two minutes of vigorous shaking in an orbital shaker. This was followed by a 10-min stabilization period on the plate before the luminescent reading. A microplate reader measured the absorbance of the mixed solutions (Molecular Devices Filter Max F5; Sunnyvale, CA, USA) [23]. One disc from each group was fixated with 4% PFA, and sputter-coating of platinum/palladium was conducted using an EMS-150T S sputter coater (Electron Microscopy Sciences, Hatfield, PA, USA). The coating thickness was selected at 5 nm. SEM in the backscattered electrons (BSE) mode analysis was completed to examine the morphology of the attached fibroblast and osteoblast on the discs at 5 kV, with a BSD4 detector, which primarily emphasizes atomic number contrast.

### 2.4. Statistical Analyses

One-way analysis of variance (ANOVA) was conducted to compare measurements between the different groups, and the Tukey HSD post hoc test was applied to assess statistically significant differences. Data that did not show normal distribution after testing for normality were analyzed using the Kruskal–Wallis non-parametric test and post hoc Dunn’s multiple comparison test. Statistical significance was determined for *p*-values less than 0.05, indicating significant differences in the data.

## 3. Results

### 3.1. Surface Modification of Machined Surface Titanium Discs after Each Treatment

After each treatment, gross examination revealed noticeable surface damage on the machined surface titanium discs when using ultrasonic and scalers, but no noticeable damage with the laser group (Figure 1a).

When compared to the control group, using 1000× magnification, the Er,Cr:YSGG laser-treated discs showed a similar appearance after treatment. Consistent results were observed in SEM analyses. Between the laser groups, the surface structure appeared to undergo fewer changes with the laser treatment at 40 Hz than at 30 Hz (Figure 1b). 

The three-dimensional profilometric surface topography showed similar results to SEM images (Figure 2a).

The mean roughness (Sq) measurements were 0.406 ± 0.03 μm, 0.373 ± 0.081 μm, 0.366 ± 0.112 μm, 0.425 ± 0.068 μm, and 0.379 ± 0.009 μm for control discs, and discs treated with titanium scalers, ultrasonic scalers, and the Er,Cr:YSGG laser at 30 Hz and 40 Hz, respectively. The mean roughness (Sa) measurements were 0.331 ± 0.030 μm, 0.293 ± 0.072 μm, 0.290 ± 0.091 μm, 0.340 ± 0.049 μm, and 0.307 ± 0.007 μm for control discs, and discs treated with hand scalers, ultrasonic scalers, and the Er,Cr:YSGG laser with 30 Hz and 40 Hz, respectively. The mean of areal material ration (Smr) values were 0.842 ± 0.308%, 13.7 ± 18.2%, 17.3 ± 14.2%, 2.19 ± 1.60%, and 1.39 ± 0.910% for control discs, and discs treated with hand scalers, ultrasonic scalers, and the Er,Cr:YSGG laser with 30 Hz and 40 Hz, respectively (Figure 2b). There was no statistical significance in Sq, Sa, and Smr among the groups (*p* > 0.05).

### 3.2. Surface Alteration of Rough Surface Titanium Discs after Each Treatment

After treatment of rough surface titanium discs, surface examination revealed marked damage to the discs from ultrasonic scalers and hand scalers, in contrast to the undamaged appearance of discs treated with the laser groups (Figure 3a).

SEM analyses also revealed noticeable changes in appearance for both laser groups. On the other hand, titanium and ultrasonic scaler treatments caused surface alteration of the rough surface with the destruction of the microstructure (Figure 3b).

The titanium and ultrasonic scaler groups tended to have less surface roughness than the others. The three-dimensional profilometric surface topography showed similar results to SEM images (Figure 4a).

The mean roughness (Sq) measurements were 2.95 ± 0.326 μm, 2.94 ± 0.343 μm, 2.35 ± 0.521 μm, 2.07 ± 0.080 μm, and 2.17 ± 0.200 μm for control discs, and discs treated with hand scalers, ultrasonic scalers, and the Er,Cr:YSGG laser with 30 Hz and 40 Hz, respectively. The mean roughness (Sa) measurements were 2.34 ± 0.253 μm, 2.37 ± 0.278 μm, 1.90 ± 0.430 μm, 1.64 ± 0.046 μm, and 1.73 ± 0.150 μm for control discs, and discs treated with hand scalers, ultrasonic scalers, and the Er,Cr:YSGG laser with 30 Hz and 40 Hz, respectively. There was no statistical significance of Sq and Sa among the groups (*p* > 0.05). The mean of areal material ration (Smr) values were 0.073 ± 0.034%, 1.12 ± 0.402%, 11.8 ± 9.28%, 0.117 ± 0.006%, and 0.134 ± 0.010% for control discs, and discs treated with hand scalers, ultrasonic scalers, and the Er,Cr:YSGG laser with 30 Hz and 40 Hz, respectively (Figure 4b). There was statistical significance in Smr between the control discs and the discs treated with ultrasonic scalers (*p* = 0.019).

### 3.3. Fibroblast Cell Attachment on Machined Surface Titanium Discs after Each Treatment

When comparing cell attachment following treatment across the four groups and the control (no treatment) at 24 h after treatments, there were no statistically significant differences in total cell count between the groups (*p* > 0.05), while the total cell count of the Er,Cr:YSGG laser with the 40 Hz group was the highest among groups in this study (Figure 5a).

Meanwhile, the cell count of the Er,Cr:YSGG laser with 40 Hz and control groups was statistically higher than that of the titanium scaler group at 72 h after treatments (*p* = 0.023 and 0.039, respectively). SEM analysis revealed that the cellular structures in the Er,Cr:YSGG laser groups exhibited a more pronounced three-dimensional morphology compared to those in the other groups (Figure 5b).

### 3.4. Osteoblast Cell Attachment on the Rough Surface Titanium Discs after Treatment

The total cell count of the Er,Cr:YSGG laser with 40 Hz group was the highest among the groups in this study, and there was a statistically significant difference between the Er,Cr:YSGG laser with 40 Hz and control groups at 24 h post-treatment (*p* = 0.045). Cell attachment comparisons among the four treatment groups and the control group showed no statistically significant differences in total cell count at 72 h post-treatment (*p* > 0.05). The total cell count of the titanium scaler group was similar to that of the control group. On the other hand, the total cell counts of the ultrasonic scaler and the Er,Cr:YSGG laser with 30 and 40 Hz were higher than those of the control group (Figure 6a).

The SEM and SEM in BSE mode images demonstrated that the interactions among cells through extended filopodia in the treatment groups with the Er,Cr:YSGG laser at 30 and 40 Hz were more than those of the control and other treatment groups, suggesting an enhanced ability of osteoblasts for attachment and intercellular interactions as a result of the Er,Cr:YSGG laser treatment (Figure 6a,b).

## 4. Discussion

Various conventional mechanical debridement techniques, including ultrasonic scalers, hand curettes, and other methods involving air abrasion devices, rotating titanium brushes, and laser devices, have been investigated to treat peri-implantitis [26,27,28,29,30,31]. However, conventional mechanical debridement can damage the microstructure of the implant surface [26,31]. Laser treatments have become widely used in many dental fields, including treatments for caries, endodontic therapy, periodontal treatment, teeth whitening, etc. [32]. The Er,Cr:YSGG lasers can decontaminate the microstructures of implant surfaces without resulting in mechanical damage [16].

In this study, the Er,Cr:YSGG laser groups with 1.5 W, an air/water ratio of 40%/50%, H mode, and 30 or 40 Hz did not exhibit any noticeable changes in the machined and rough titanium surfaces, and no statistically significant differences were observed between the two laser groups. These results were consistent with those of a previous investigation [33]. On the other hand, the ultrasonic and titanium scalers caused notable changes in the machined and rough titanium surfaces, which were consistent with the results from previous studies [16,34]. A statistically significant difference was observed in the Smr value of the ultrasonic scalers group compared to the control group using the rough surface titanium discs, strongly suggesting that caution should be exercised when using an ultrasonic scaler on rough surfaces. Notably, in the results of machined surface titanium discs, the Er,Cr:YSGG laser under 30 Hz resulted in a more uneven surface compared to discs treated with 40 Hz lasers. In contrast, on rough surface titanium discs, the 40 Hz laser group resulted in a more uneven surface compared to the 30 Hz laser group. A possible explanation for this discrepancy is that despite the laser having a stronger irradiation potential to melt the surface of discs at 30 Hz, the type of surface may also be contributory. On machined surface titanium discs, the surfaces were inherently smooth before any treatment. Using the laser at 30 Hz would create concavities that make the surface less even. On rough surface titanium discs, the surfaces were already irregular at baseline. The usage of a laser at 30 Hz would melt the surface and actually decrease the irregularities to create a more even surface. Despite these findings, it is important to note that there were no statistically significant differences.

Fibroblast cells are responsible for producing collagen fibers and are found in the connective tissue surrounding peri-implant areas, predominantly located adjacent to the surfaces of implant abutments [35]. Fibroblast cells play a crucial role in forming a soft tissue seal around implants and have been extensively studied in vitro for their attachment, spreading, and proliferation [36,37]. Past in vitro studies reported that conventional mechanical debridement can damage the implant surface’s microstructure and reduce cell activity on the compromised microstructured surface [27,34]. However, the present study showed that the number of fibroblasts on the machined surface was higher than in the other groups when treated with the Er,Cr:YSGG laser groups with 40 Hz at 24 h. Similarly, an in vitro study using the Er,Cr:YSGG laser and human gingival fibroblast cells reported that the Er,Cr:YSGG laser significantly increased human gingival fibroblast cell proliferation compared to the no-treatment group [38]. Another in vitro study reported that the Er,Cr:YSGG laser-treated titanium disc surfaces significantly increased fibroblast viability at Days 1, 3, and 5 compared to the no-treatment group using the following parameters: 1.5 W, 20 Hz, with 11% distilled water and 7% air for 30 s. At 72 h in this study, the total fibroblast cell count of the Er,Cr:YSGG laser groups with 40 Hz was similar to that of the no-treatment group. The discrepancy may be due to the differences in the laser irradiation conditions, and further consideration of laser irradiation conditions might be recommended in future studies.

Microrough surfaces offer the benefits of enhanced bone-implant contact and osseointegration [39,40]. They also lead to improved mechanical interlocking with bone compared to smoother surfaces such as machined surfaces [41]. The Er,Cr:YSGG laser irradiation group exhibited greater and earlier osteoblast cell spreading on the zirconia surface compared to the non-irradiated group [42]. In this study, the osteoblast proliferation and attachment in the Er,Cr:YSGG laser groups with 40 Hz were statistically higher than those in the control group. The Er,Cr:YSGG laser groups may have an effect on increasing osteoblast proliferation and attachment on dental implant surfaces. Surprisingly, the ultrasonic scaler group in this study showed the highest osteoblast cell proliferation on rough titanium surfaces among the other groups in this study. The osteoblast culture on a micro rough surface demonstrated decreased osteoblast attachment and slower proliferation but faster differentiation than on the machined surface [43]. The result of the ultrasonic scaler group in this study might be due to changes in morphology of rough titanium surfaces to smoother surfaces similar to machined titanium surfaces. The Er,Cr:YSGG laser groups with 30 Hz and 40 Hz showed a higher total cell number of osteoblast cells on the rough titanium surface than that of the control groups. An in vitro study by Romanos et al. 2006 reported that the Er,Cr:YSGG laser irradiation of the titanium surface presented a higher cellular density than the no-treatment group, and pseudopodia and a spread of cells, which demonstrated maturation were observed in the Er,Cr:YSGG laser group [44]. In this study, the interactions among cells through extended filopodia in the Er,Cr:YSGG laser with 30 and 40 Hz groups were observed. The Er,Cr:YSGG laser irradiation of titanium surfaces can enhance osteoblast attachment and subsequently promote bone formation.

The limitations of this study are as follows: first, only two laser irradiation conditions were used; second, the sample size was small, which may affect the statistical power and generalizability of the results; third, only cell proliferation was observed without differentiation. Even though the Er,Cr:YSGG laser was used, changes in the roughness parameters were observed with no visible changes in the surface properties. Also, not all proliferating cells are differentiated. Therefore, future research should explore various laser irradiation conditions, increase the sample size, and utilize RNA sequencing to observe gene expression, including cell differentiation markers. However, this study could show the benefits of using the Er,Cr:YSGG laser for peri-implantitis.

## 5. Conclusions

This study demonstrated how dental implant surface changes when using ultrasonic and titanium scalers compared to using the Er,Cr:YSGG laser. The Er,Cr:YSGG laser effectively promoted fibroblasts and osteoblasts to adhere to the titanium surface without significant surface changes, suggesting that the Er,Cr:YSGG laser might be superior compared to other debridement methods for the peri-implantitis treatment.

## Figures and Tables

**Figure 1 materials-17-04899-f001:**
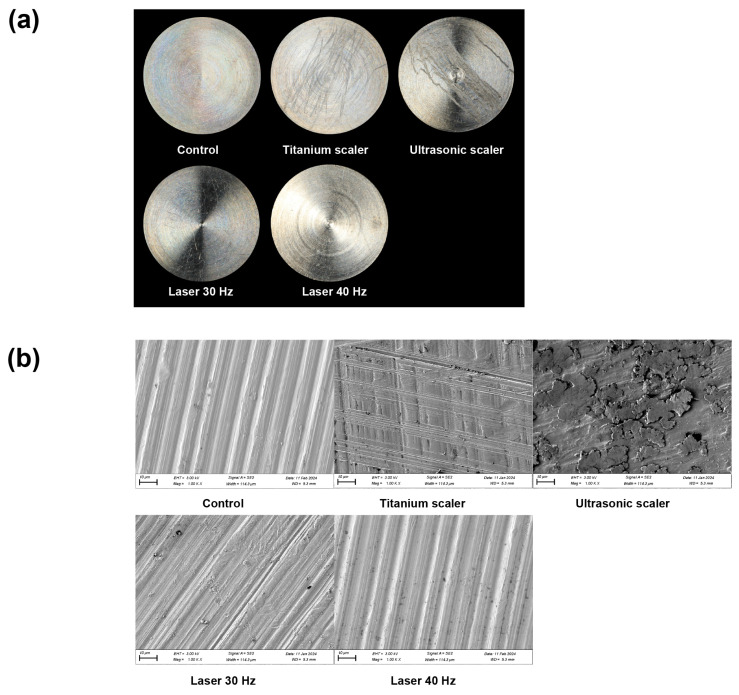
Machined surface characterization following different treatments. (**a**) Representative images of the machined surface after each treatment. (**b**) SEM analysis of each disc after treatment at 1000× magnification.

**Figure 2 materials-17-04899-f002:**
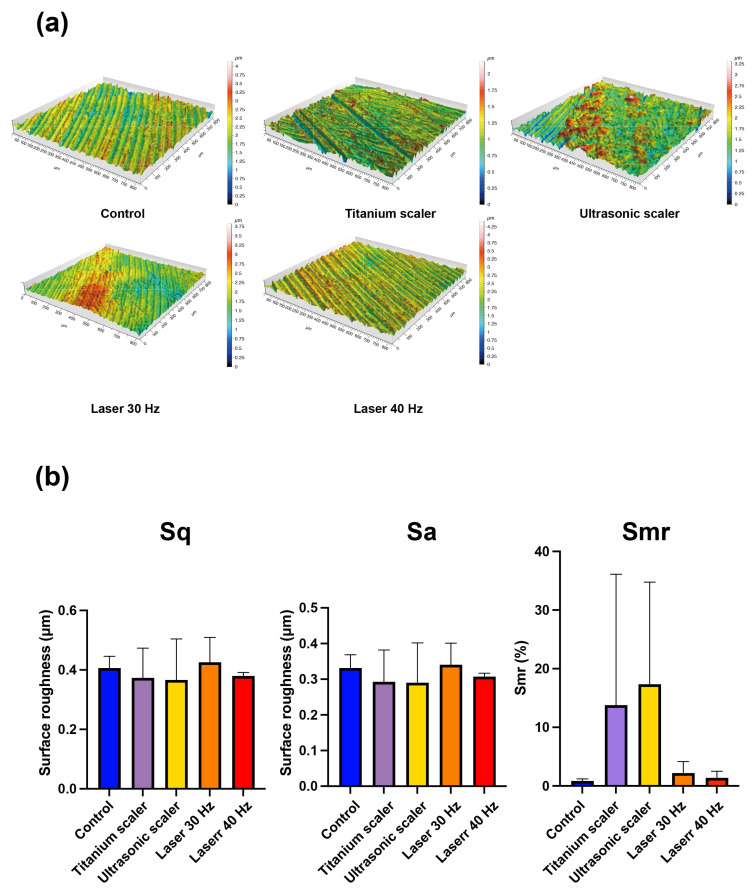
Characterization of machined surfaces following various treatments was measured using an optical profilometer. (**a**) Surface roughness parameters, including squared mean height (Sq), arithmetic mean height (Sa), and the areal material ration (Smr). (**b**) 3D topographic images of each disc after treatment.

**Figure 3 materials-17-04899-f003:**
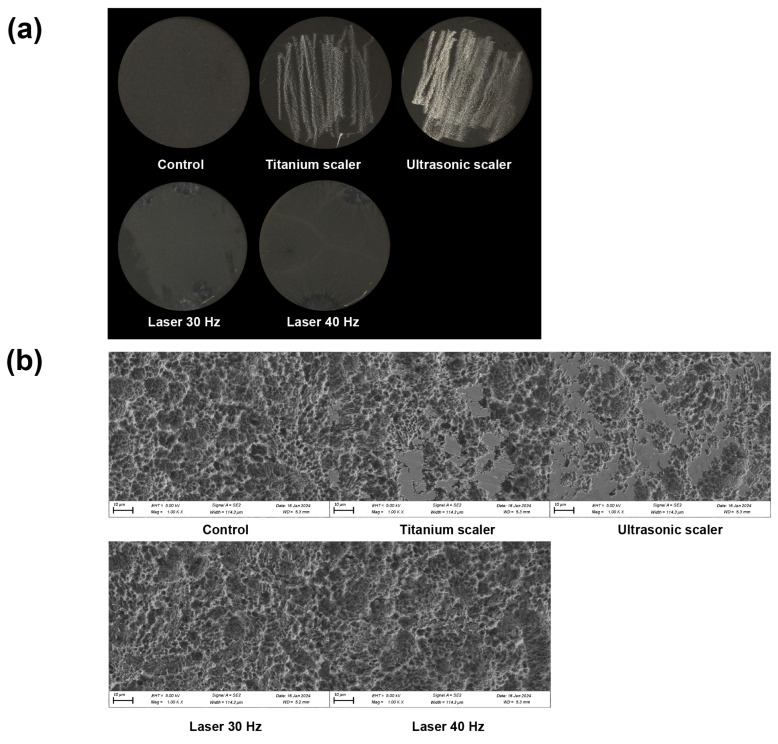
Characterization of rough surfaces after various treatments: (**a**) Representative images of each rough surface post-treatment, (**b**) SEM analysis of each disc following treatment at 1000× magnification.

**Figure 4 materials-17-04899-f004:**
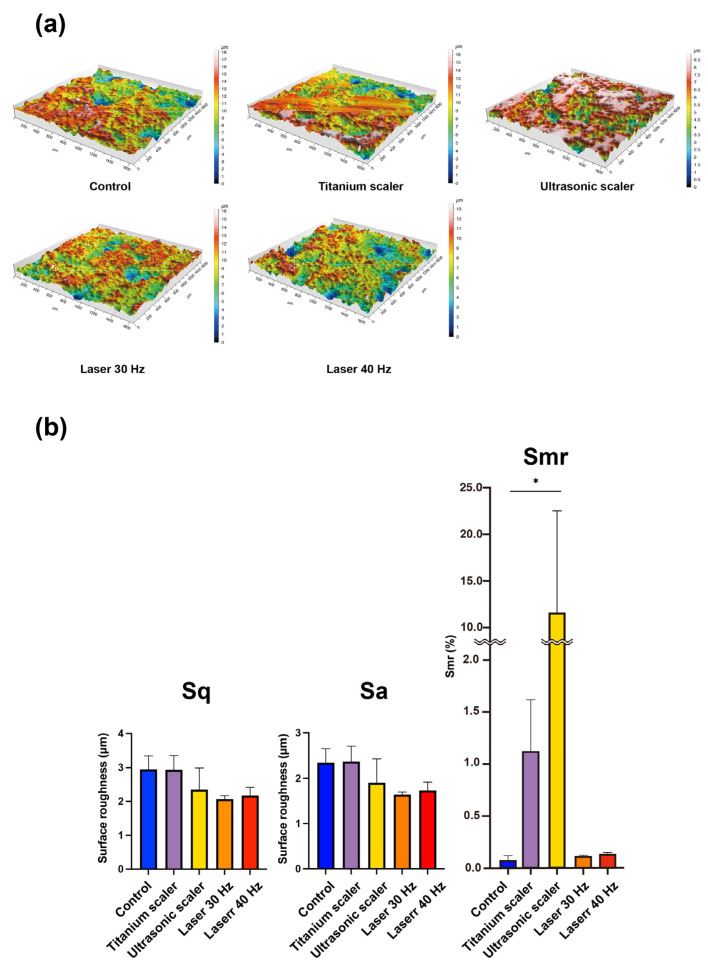
Characterization of rough surfaces following various treatments was measured using an optical profilometer. (**a**) 3D topographic images of each disc after treatment. (**b**) Surface roughness parameters, including Sq (squared mean height), Sa (arithmetic mean height), and Smr (areal material ration) (* *p* < 0.05).

**Figure 5 materials-17-04899-f005:**
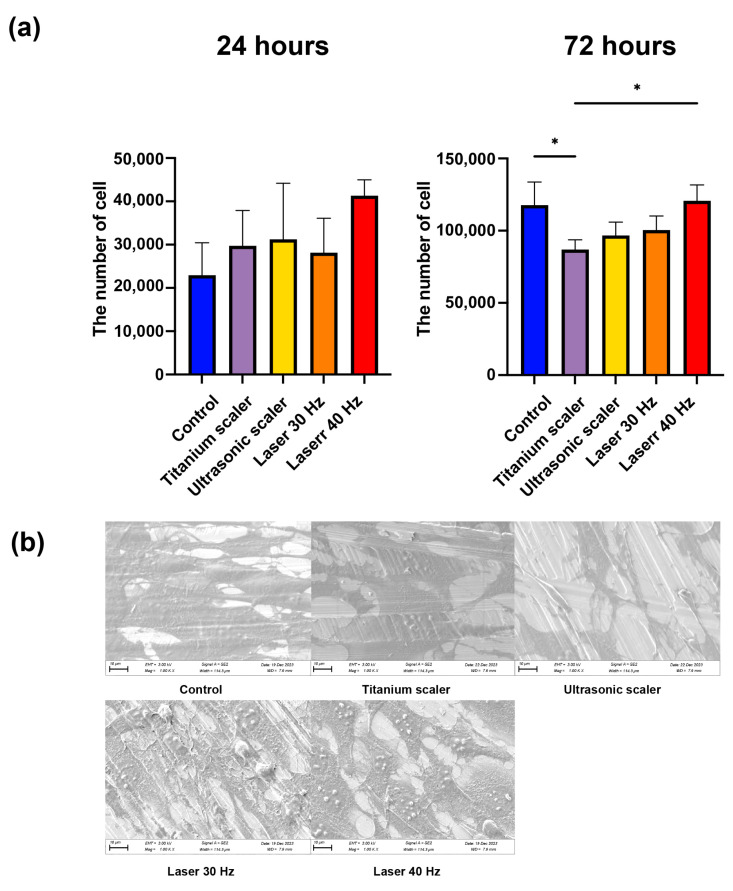
Fibroblast cell attachment following surface treatment. (**a**) Cell count of fibroblasts 24 and 72 h after seeding (* *p* < 0.05). (**b**) SEM analysis of each disc and fibroblasts at 1000× magnification.

**Figure 6 materials-17-04899-f006:**
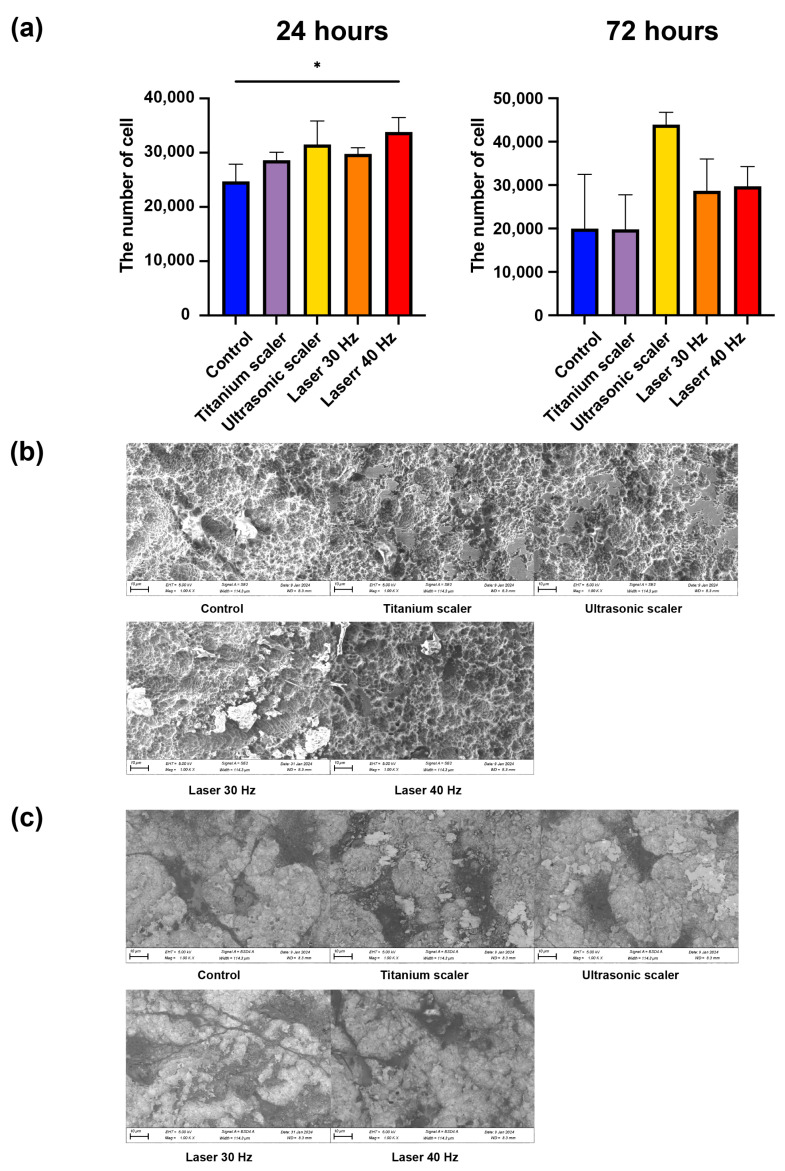
Osteoblast cell attachment following surface treatment (* *p* < 0.05). (**a**) Cell count of osteoblasts 24 and 72 h after seeding. (**b**) SEM analysis of each disc and osteoblast at 1000× magnification. (**c**) SEM in BSE mode analysis of each disc and osteoblast at 1000× magnification.

## Data Availability

All data are included in the manuscript.

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
