# Peer review of "Effect of Er,Cr:YSGG Laser Irradiation on the Surface Modification and Cell Adhesion on Titanium Discs: An In Vitro Study"

_materials, 2024, doi:10.3390/ma17194899_

Round 1
Reviewer 1 Report
Comments and Suggestions for Authors
The manuscript entitled ” Effect of Er,Cr:YSGG Laser Irradiation on the Surface Modification and Cell Adhesion on Titanium Discs: an In Vitro Study” by Takahiko Shiba, Kailing Ho, Xuehao Ma, Ye Won Cho, Chia-Yu Chen, and David M. Kim, highlights a comparative study of the cell adhesion on Ti samples with different roughness values. The authors have inserted many results, but my suggestion is to clarify the used protocol of the Ti sample treatments. The paper can be published after Major changes. My comments/ suggestions are below.
1. Introduction section: the authors are kindly asked to insert the purpose of the conducted experiments, along with the novelty of their research approach.
2. Line 82: The authors said, “Smooth and rough surface…” What do you mean by “smooth”? By polishing the Titanium samples with diamond polishing paste, you can obtain surfaces with roughness below 90 nm (this statement is made by my own lab. experience). Is there a reason that you are using surface roughness in the micron values domain?
3. Line 94: “…and 5. Laser-treated instrumentation was done using an Er,Cr:YSGG laser…”. Please reformulate it; what is the purpose for starting a new line with “and.5.”, when you have to insert the information for bullet 4?
4. Figure 1: The graph concerning the Atomic percentage of Oxygen seems to present a higher amount of Oxygen for the case when the sample was prepared by ultrasonic method. My concern is that the Oxygen content is coming from the preparation method, and it should not be taken into account. What are the opinions of the authors?
5. Figure 2: Concerning the roughness of the Titanium materials. It seems that in the case of the 30 Hz Laser, the roughness is higher than in the control samples, and the 40 Hz Laser seems to offer results approximatively the same as the Control sample. Why does the 30Hz Laser increase the surface roughnesses instead of decreasing (by melting) them? What is the scientific mechanism?
6. Please explain what is the difference between sections 3.1 and 3.2. In section 3.1, entitled “Surface modification of the machine surface,” you present Ti materials exposed to the Laser, and in section 3.2, entitled “Surface alteration of the rough surface,” you present the same samples that were treated, but it is not explained how. Also, on two of the samples (in Figure 3 a), appears some big scratches, which suggests that they were made manually by manipulating a rough material. Please reformulate the section clearly.
Author Response
Reviewer #1 (Comments for the Author):
The manuscript entitled “Effect of Er,Cr:YSGG Laser Irradiation on the Surface Modification and Cell Adhesion on Titanium Discs: an In Vitro Study” by Takahiko Shiba, Kailing Ho, Xuehao Ma, Ye Won Cho, Chia-Yu Chen, and David M. Kim, highlights a comparative study of the cell adhesion on Ti samples with different roughness values. The authors have inserted many results, but my suggestion is to clarify the used protocol of the Ti sample treatments. The paper can be published after Major changes. My comments/ suggestions are below.
Response: We would like to thank you for reviewing our manuscript. We carefully revised the manuscript based on your valuable feedback.
- Introduction section: the authors are kindly asked to insert the purpose of the conducted experiments, along with the novelty of their research approach.
Response: Thank you for your suggestion. We have added our experimental purpose and novelty as follows on lines 75-84 in the revised manuscript. Only a few papers reported the effect of Er,Cr:YSGG laser on both machined and rough surfaces. Furthermore, to the best of our knowledge, no publications have concurrently assessed the proliferation rates of fibroblasts and osteoblasts after laser irradiation of both surfaces compared to other methods.
It is essential to consider appropriate laser irradiation conditions for treating peri-implantitis on machined and rough surfaces and for cell proliferation of fibroblast and osteoblast after the treatment. The purpose of the present study was (a) to compare Er,Cr:YSGG laser-induced changes in machined and rough titanium surfaces with mechanical methods; (b) to evaluate the effect of Er,Cr:YSGG laser on fibroblast and osteoblast cell attachment and morphology on machined surface and rough surface titanium discs.
- Line 82: The authors said, “Smooth and rough surface…” What do you mean by “smooth”? By polishing the Titanium samples with diamond polishing paste, you can obtain surfaces with roughness below 90 nm (this statement is made by my own lab. experience). Is there a reason that you are using surface roughness in the micron values domain?
Response: Thank you for your suggestion. We apologize for the confusion. We have changed from “smooth” to “machined” in all applicable locations.
- Line 94: “…and 5. Laser-treated instrumentation was done using an Er,Cr:YSGG laser…”. Please reformulate it; what is the purpose for starting a new line with “and.5.”, when you have to insert the information for bullet 4?
Response: Thank you for your suggestion. We have reformulated and separated groups 4 and 5 as follows on lines 100-108 in the revised manuscript:
- Laser-treated instrumentation was done using an Er,Cr:YSGG laser (Waterlase iPlus, Biolase, Irvine, CA, USA) with a wavelength of 2780 nm in short pulse “H” mode and 60 μs using a radial firing tip (RFPT 5) with a spot size of 1 mm. The power setting was 1.5 W with air/water ratio of 40%/50%. The pulse rate was 30 Hz [23]. The laser tip was affixed to a handle and manually moved back and forth across the surface while maintaining a constant speed and distance of about 0.5 mm between the tip and the surface.
- In this laser-treated group, the pulse irradiation condition of group 4 was changed to 40 Hz, while all other conditions remained the same.
- Figure 1: The graph concerning the Atomic percentage of Oxygen seems to present a higher amount of Oxygen for the case when the sample was prepared by ultrasonic method. My concern is that the Oxygen content is coming from the preparation method, and it should not be taken into account. What are the opinions of the authors?
Response: Thank you for your comment. We observed this phenomenon in both machined and rough surfaces. We believe that the cause of higher oxygen levels on the titanium surface in the ultrasonic group on both machined and rough surfaces is due to heat generation. Consequently, the titanium content in the ultrasonic group may be reduced titanium content in these surfaces. We commented on the discussion section as follows on lines 275-283 in the revised manuscript: Interestingly, the amount of oxygen on the titanium surface was elevated in the ultrasonic group compared to the other groups in machined and rough surfaces. It was reported that thicker oxide layers on titanium Grade Ⅱ, varying from 1.53 ± 0.24 to 6.01 ± 0.22 mm, were produced via oxidation at 700°C for durations ranging from 6 to 72 hours. The hardness of these oxide layers increased with both higher temperatures and longer oxidation times [23]. In this study, it is estimated that a considerable amount of heat was generated on the titanium surface in the ultrasonic group. Consequent to the result, it was presumed that the amount of titanium element in the machine and rough surfaces decreased in the ultrasonic group.
- Figure 2: Concerning the roughness of the Titanium materials. It seems that in the case of the 30 Hz Laser, the roughness is higher than in the control samples, and the 40 Hz Laser seems to offer results approximatively the same as the Control sample. Why does the 30Hz Laser increase the surface roughnesses instead of decreasing (by melting) them? What is the scientific mechanism?
Response: Thank you for your comment. As you pointed out, the laser with 30 Hz group tended to have a rougher surface than the laser with 40 Hz from the result of a titanium machined surface. On the other hand, the laser with 40 Hz group tended to have a rougher surface than the laser with 30 Hz from the result of a titanium rough surface. As you mentioned, 30 Hz is more likely to cause surface melting than 40 Hz. Therefore, the roughness of titanium rough surfaces in the laser 30 Hz groups was smoother than that of the laser with 40 Hz. The machined surface had regular irregularities. The concave areas in the laser with 30HZ were more melted than the laser with 40 Hz group, as can be seen in the SEM images of the titanium machined surface. The results of the titanium machined surface showed the rougher surface roughness at the laser with the 30Hz group than that of the laser with 40Hz, although there is no statistical difference. We have added these findings to the discussion section as follows on lines 286-297 in the revised manuscript: Notably, in the results of machined surface titanium discs, the Er,Cr:YSGG laser under 30 Hz resulted in a more uneven surface compared to discs treated with 40 Hz lasers. In contrast, for rough surface titanium discs, the 40 Hz laser group resulted in a more uneven surface compared to the 30Hz laser group. A possible explanation for this discrepancy is that despite the 30Hz laser having a stronger irradiation potential to melt the surface of discs, the type of surface also affects the end result. In machined surface titanium discs, the surfaces were smooth to begin with. Using a 30Hz laser would create concavities that end up making the surface less even. With rough surface titanium discs, the surfaces were already irregular to begin with. Using a 30Hz laser would melt the surface and actually decrease the irregularities to create a more even surface. Despite these findings, it is important to note that there were no statistically significant differences.
- Please explain what is the difference between sections 3.1 and 3.2. In section 3.1, entitled “Surface modification of the machine surface,” you present Ti materials exposed to the Laser, and in section 3.2, entitled “Surface alteration of the rough surface,” you present the same samples that were treated, but it is not explained how. Also, on two of the samples (in Figure 3 a), appears some big scratches, which suggests that they were made manually by manipulating a rough material. Please reformulate the section clearly.
Response: Thank you for your comment. We have shown the results of machined and rough surface titanium disks in sections 3.1 and 3.2, respectively. As per your suggestion, we have revised each subtitle and some sentences in sections 3.1 and 3.2 to avoid confusion on lines 158-195 in the revised manuscript. The current subtitles of sections 3.1 and 3.2 are “Surface modification of machined surface titanium disks after each treatment” and “Surface alteration of rough surface titanium disks after each treatment,” respectively.

Reviewer 2 Report
Comments and Suggestions for Authors
This experimental investigation is a paper that presents information for researchers in the field of cell adhesion in implant surfaces.
Abstract
This section must report each section of the paper.
Introduction
The introduction shows a state of art of decontamination aspects for decontamination of titanium surfaces in implant dentistry.
Different mechanical and chemical decontamination approaches, such as the usage of ultrasonic scalers, titanium curettes, plastic curettes, erbium-doped yttrium aluminium garnet (Er:YAG) laser, and erbium, chromium-doped:yttrium, scandium, gallium, and garnet (Er,Cr:YSGG) laser have been proposed.
This study aimed to investigate the effect of Er,Cr:YSGG laser in titanium surfaces and on fibroblast and osteoblast cell attachment.
The section must be updated with recent references.
Material and methods.
This section is well structured. The authors reported several subsections (titanium discs preparation, surface characterization, fibroblast and osteoblast cell adhesion) for to explain the methodology of this experimental study.
However, the authors must to report specific references for each subsection for to explain the scientific basis of each experimental procedure.
Results. This section is well structured. The paragraphs are correct and showed the main aspects of experimental findings.
However, some figures must be improved. Figure 1a and b, Figure 2a, Figure 3b and c; Figure 5b.
Discussion. This section shows the most relevant aspects of the study with compared studies for the future researchers.
The section must be updated with recent references.
Conclusions. This section be revised. The study assesses several methods for decontamination of surface implants and the cell adhesion of fibroblasts and osteoblast. Does not include aspects of peri-implantitis treatment.
All references must be revised because only 25% are less than 5 years.
Author Response
Reviewer #2 (Comments for the Author):
This experimental investigation is a paper that presents information for researchers in the field of cell adhesion in implant surfaces.
Response: We would like to thank you for reviewing our manuscript. We carefully revised the manuscript based on your valuable feedback.
Abstract
This section must report each section of the paper.
Response: We sincerely appreciate your comment. Unfortunately, the abstract section has a character limit of 200 characters. We believe our abstract includes the important components of each section of our manuscript. Additionally, the journal requires an unstructured abstract according to its guidelines.
Introduction
The introduction shows a state of art of decontamination aspects for decontamination of titanium surfaces in implant dentistry. Different mechanical and chemical decontamination approaches, such as the usage of ultrasonic scalers, titanium curettes, plastic curettes, erbium-doped yttrium aluminium garnet (Er:YAG) laser, and erbium, chromium-doped:yttrium, scandium, gallium, and garnet (Er,Cr:YSGG) laser have been proposed. This study aimed to investigate the effect of Er,Cr:YSGG laser in titanium surfaces and on fibroblast and osteoblast cell attachment. The section must be updated with recent references.
Response: Thank you for your suggestions. We have updated them to recent references.
Material and methods
This section is well structured. The authors reported several subsections (titanium discs preparation, surface characterization, fibroblast and osteoblast cell adhesion) for to explain the methodology of this experimental study. However, the authors must to report specific references for each subsection for to explain the scientific basis of each experimental procedure.
Response: Thank you for your suggestions. We have added as many references as possible.
Results
This section is well structured. The paragraphs are correct and showed the main aspects of experimental findings. However, some figures must be improved. Figure 1a and b, Figure 2a, Figure 3b and c; Figure 5b.
Response: I sincerely appreciate your suggestions. We understand that you pointed
out the resolution of these figures. According to the journal’s guidelines, we have uploaded
higher-resolution images to the edited version.
Discussion
This section shows the most relevant aspects of the study with compared studies for the future researchers. The section must be updated with recent references.
Response: Thank you for your suggestions. We have updated them to recent references.
Conclusion
This section be revised. The study assesses several methods for decontamination of surface implants and the cell adhesion of fibroblasts and osteoblast. Does not include aspects of peri-implantitis treatment.
Response: Thank you for your suggestions. We have edited our arguments for peri-implantitis treatment in the conclusion section as follows on lines 350-353 in the revised manuscript: The Er,Cr:YSGG laser effectively promoted fibroblasts and osteoblasts to adhere to the titanium surface without significant surface changes, suggesting the Er,Cr:YSGG laser might besuperior compared to other debridement methods for the peri-implantitis treatment.
All references must be revised because only 25% are less than 5 years.
Response: Thank you for your suggestions. We have updated them to recent references as much as possible.

Round 2
Reviewer 1 Report
Comments and Suggestions for Authors
The authors have responded punctually to all Reviewer’s suggestions and comments, improving the quality of the present manuscript. The paper can be accepted for publication.
Reviewer 2 Report
Comments and Suggestions for Authors
The review is correct